# An Iterative Learning Scheme with Binary Classifier for Improved Event Detection in Surveillance Video

**Cuong H. Tran and Seong G. Kong ***

Department of Computer Engineering, Sejong University, Seoul 05006, Republic of Korea; tdhcuong@sju.ac.kr
* Correspondence: skong@sejong.edu

**Abstract:** This paper presents an iterative training framework with a binary classifier to improve the learning capability of a deep learning model for detecting abnormal behaviors in surveillance video. When a deep learning model trained on data from one surveillance video is deployed to monitor another video stream, its abnormal behavior detection performance often decreases significantly. To ensure the desired performance in new environments, the deep learning model needs to be retrained with additional training data from the new video stream. Iterative training requires manual annotation of the additional training data during the fine-tuning process, which is a tedious and error-prone task. To address this issue, this paper proposes a binary classifier to automatically label false positive data without human intervention. The binary classifier is trained on bounding boxes extracted from the detection model to identify which boxes are true positives or false positives. The proposed learning framework incrementally enhances the performance of the deep learning model for detecting abnormal behaviors in a surveillance video stream through repeated iterative learning cycles. Experimental results demonstrate that the accuracy of the detection model increases from 0.35 (mAP = 0.74) to 0.91 (mAP = 0.99) in just a few iterations.

**Keywords:** iterative learning; binary classifier; event detection; surveillance video; deep learning

## 1. Introduction

Detecting abnormal behaviors from a video stream has drawn significant attention in the computer vision and machine learning research community. Abnormal behaviors can be defined as events deviating from ordinary behaviors [1]. Abnormal behavior detection can be useful in detecting unusual behaviors of animals, such as the estrus habits of cows or cribbing of horses. In livestock farming, cattle estrus detection is the most crucial factor in monitoring cattle health and breeding management. The most obvious signal is the mounting behavior in which a cow mounts another cow for a short duration. Failing to detect the estrus behaviors will not only reduce the conception rates for the next generation but also milk production. The process of detecting abnormal events in videos requires the labor-intensive attention of human operators since abnormal events happen only 0.1% of the time and 99.9% of watching is wasted [2]. Especially, the barn can contain hundreds of cows which makes the monitoring process costly and error-prone. Intelligent surveillance monitoring systems should ideally detect abnormal events automatically in streaming video in real time. While breeding can be managed under controlled conditions, accurately tracking the estrus behaviors of cows is essential to determine the optimal timing for breeding. Estrus events occur infrequently, making continuous monitoring with surveillance cameras indispensable to identify the most suitable period for successful breeding.

Existing work on abnormal behavior detection in surveillance videos can be categorized into traditional and deep learning-based methods. Traditional approaches rely on hand-crafted features, which require high computational costs to estimate, and the detection performance may decrease significantly when deployed to monitor scenes with cluttered backgrounds. Deep learning-based approaches for abnormal behavior detection often

employ fast object detection models such as You Only Look Once version 5 (YOLOv5) [3] to learn the features to characterize abnormal behaviors [4–6]. Detection models with high processing speeds and reasonably high detection accuracies can be implemented on edge computing devices such as Jetson Nano. However, a challenge arises when dealing with background information, including noise and irrelevant moving objects. Treating this background information as additional features can hinder the object detection model from learning the prevalent features that characterize abnormal behaviors effectively. Moreover, when deploying an object detection model pre-trained on a specific video stream to monitor a different surveillance video stream, the model is likely to produce a significant number of false positives and false negatives. This issue arises because the model was originally pre-trained to learn the features of abnormal behaviors based on the specific backgrounds used during its training, which may not align with the new surveillance environment. Therefore, the model's performance decreases significantly when compared to its performance on a video stream with similar backgrounds to those in the training dataset. To address this issue, the detection model needs to undergo retraining using data specifically collected from the new video streams. This will enable the model to better adapt to the characteristics of the changing environments, ultimately improving its detection performance. To ensure comparable performance in the new video stream, the base detection model must be retrained using additional training data acquired from the new video stream. However, collecting and labeling such data is a time-consuming process that requires significant human effort. Once the data is labeled, the detection model can be fine-tuned by adjusting its internal parameters, enabling it to better detect abnormal behaviors in the new operating environment.

This paper proposes an iterative training framework that employs a binary classifier to enhance the learning capability of a deep learning model for detecting abnormal behaviors in surveillance videos while reducing the number of false positives. The iterative learning process involves repeating the training in a few cycles to gradually improve the performance of the detection model using the base training dataset. The base detection model is trained using a dataset that includes the manually labeled bounding boxes of objects exhibiting abnormal behaviors from surveillance video streams in both daytime and nighttime scenarios. Subsequently, the trained base model is deployed to a new surveillance video to monitor the same abnormal events. However, a notable decrease in the detection performance of abnormal behaviors is observed when the video stream comes from an environment significantly different from the scenes used to train the base model. To address this issue, the detection results are monitored over a certain time period to obtain both false positive (FP) and true positive (TP) images. Only the TP results are added to the existing dataset, which then becomes the training dataset for the next training cycle. The collected TP images need to be appropriately labeled for another supervised learning process in the next training cycle. In this study, a base binary classifier is trained to correct the labels of FP images, which contain incorrect predictions of abnormal objects from the detection model. After obtaining the new corrected training dataset, the trained base detection model is fine-tuned to adapt its weights to the current environment. The bounding boxes corresponding to TP and FP objects are subsequently collected to train a new binary classifier for the next training cycle. The fine-tuned model is then deployed to the same environment, and the streaming is monitored for a certain time duration. If the performance does not meet the target, another training cycle with the fine-tuned detection model and a new binary classifier using the same processing steps is repeated until the desired performance is achieved. This iterative approach aims to gradually improve the model's performance and adapt it to the specific environment in which it is deployed.

The main contributions of this paper are summarized as follows: (1) the proposal of an iterative learning scheme for a scalable deep learning model that can detect abnormal behaviors from surveillance videos in a new environment and (2) the utilization of a binary classifier in the iterative training process which reduces the need for human efforts to label the streaming data collected from the changing environment. The remainder of this paper is

organized as follows: Section 2 describes related works on abnormal behavior detection in surveillance video streams. Section 3 explains the details of our proposed iterative learning scheme with a binary classifier for detecting abnormal behaviors. Section 4 presents the experiment results, evaluation, and performance evaluation. Section 5 discusses ablation studies for our approach. Finally, Section 6 concludes the paper.

## 2. Related Work

Abnormal behavior detection is a well-established problem in the field of computer vision. The main objective is to identify actions occurring in a video and analyze abnormal events. However, a significant challenge in abnormal behavior detection arises from the variability of actions based on the specific environment. To address this issue, a common approach is to efficiently extract features from image sequences. These features are then utilized for various tasks such as object detection, pose estimation, and dense trajectories. Traditionally, machine learning algorithms have shown impressive results in detecting abnormal human behavior by employing shallow feature learning from video data. Methods like random forest (RF) [7], Bayesian networks [8], Markov models [9], and support vector machine (SVM) [10] have been used to recognize human behaviors. However, these methods heavily rely on pre-processing and handcrafted features, which require a significant amount of time and resources to process. Consequently, they do not scale well to different datasets and exhibit poor performance in real-world scenarios [11]. Deep learning methods, however, have gained considerable interest from the community due to their ability to automatically extract meaningful features. Unlike traditional machine learning methods, deep learning involves a multi-stage learning process that automatically extracts representative features for a specific task through several hidden layers [12]. These features, known as deep features, exhibit scalability across various scenarios. Deep learning has recently been applied in abnormal behavior detection and has proven to be highly efficient in video surveillance systems [13]. There are several different approaches to abnormal behavior detection: (i) handcrafted feature-based abnormal behavior detection, (ii) sparse coding-based abnormal behavior detection, and (iii) abnormal behavior detection using end-to-end deep learning models.

Handcrafted feature-based abnormal behavior detection involves manually extracting features from a scene to detect abnormal events. Tung et al. [14] utilized low-level trajectories for detection, but this method lacks robustness in crowded scenes. In [15], researchers improved this approach by incorporating the Histogram of Oriented Gradient (HOG) feature. Adam et al. [16] utilized flow features to establish the exponential distribution and model the normal behavior from the training dataset. However, handcrafted feature methods are computationally expensive and are not robust to noise and cluttered backgrounds. Sparse coding-based methods assume that normal vector features can be represented as a linear combination of basis vectors from a learning dictionary. These methods aim to find a set of basis vectors to construct the sparse representation of the normal training data, using only normal data for this purpose. In [17], the authors built a sparse coding dictionary to record only normal events and employed a large reconstruction error during inference to detect abnormal events. However, the sparse coding process is slow and time-consuming. To address this issue, Lu et al. [18] accelerated the sparse learning process by discarding the sparse regularization and proposed the use of multiple dictionaries to learn a normal distribution.

Most of the existing works on deep learning-based methods assume that normal behavior patterns can be well reconstructed or predicted. Hansen et al. [19] employed a convolutional autoencoder for reconstruction tasks and suggested stacking multiple consecutive frames in channels to improve performance. Liu et al. [20] introduced a baseline model that detects abnormal behaviors using both appearance features and optical flow for frame prediction. Luo et al. [21] proposed a sparse coding approach for anomaly detection based on the stacked RNN framework. Other abnormal behavior detection methods [4–6,22] rely on fast detection models like YOLO. Fang et al. [4] proposed an

improved architecture for YOLOv3 to detect abnormal behavior during an examination. The authors suggested the frame-alternate dual-thread method to increase the speed and accuracy of the detection model [5] and used YOLOv3 along with a Gaussian background model to generate multiple kernel learning packages to detect highway accidents. Ji et al. [6] suggested the T-TINY-YOLO model, which is based on zero-valued weight parameters to enhance the real-time performance of abnormal human behavior detection on NVIDIA Jetson TX2. Li and Dai [22] used a weighted convolutional autoencoder (Conv-AE) along with YOLOv3 to learn the regularity score for a crowded scene. Table 1 summarizes the advantages and disadvantages of different approaches for abnormal behavior detection.

**Table 1.** Summary of various approaches for detecting abnormal events in video surveillance.

| Approach | Method | Advantages | Disadvantages |
|---|---|---|---|
| Handcrafted feature-based | Low-level trajectories [14] HOG [15] Flow features [16] | ■ Simpler to implement and interpret ■ Works well with a small dataset | ■ Computationally expensive ■ Not robust to noise and cluttered backgrounds |
| Sparse coding-based | Normal sparse coding dictionary [17,18] | ■ Improved interpretability ■ Provides an efficient representation of high-dimensional data | ■ Computationally expensive ■ Limited scalability with large-scale datasets |
| Deep learning based | AutoEncoder [19] GANs [20] Stacked RNN [21] YOLO [4–6,22] | ■ End-to-end learning ■ Scalability to large and high-dimensional datasets ■ Well-suited for transfer learning | ■ Requires large amounts of training and test data ■ Lack of interpretability ■ Prone to overfitting ■ Lack of adaptiveness when deploying in new environments |
| Proposed | Iterative training with binary classifier | ■ Compatible with various object detection models ■ Adapts to new environments to reduce false positives ■ Reduces time and effort to annotate new training data | ■ Requires a few iterations for the model to adapt to a new environment |

## 3. Iterative Training of the Detection Model with a Binary Classifier

### 3.1. Iterative Training Process

The detection model for abnormal behavior heavily relies on the collected training dataset. When deployed in unfamiliar environments with diverse backgrounds and different illumination conditions, the overall performance rapidly decreases. Therefore, data-driven detection systems should be built based on the new environment. This paper introduces a new vision-based detection system that utilizes fast object detectors such as YOLOv5 [3] to detect abnormal behaviors. The system can be deployed in different environments to collect new training data. Afterward, iterative training is conducted on this new data to fine-tune the detector. A binary classifier is also employed to reduce the human effort required for labeling the collected streaming data.

The proposed iterative training scheme involves training a base deep learning model iteratively using additional training data samples collected from a new video stream. When a detection model, pre-trained on a specific dataset, is deployed to monitor a new surveillance video stream, it may produce a significant number of false positives due to differences in the environments. To improve the performance of the pre-trained detection model on a new video stream, the model needs to be retrained using the data obtained

from similar environments. The iterative training process collects and classifies positive images into true positives and false positives to correct the labels. The additional collected data is then used to fine-tune the existing detection model, making it adapt to different environments. This strategy allows for the detection model to quickly adjust its weights to the specific conditions and detect abnormal events in new video streams.

Two separate datasets, DB-I and DB-O, are used to train the detection model (DM) and the binary classifier (BC), respectively. DB-I is a dataset of images containing both normal and abnormal events which is used to train the detection model in a supervised manner. Normal images contain all normal objects, while an abnormal image contains at least one abnormal event in the scene. The object dataset (DB-O) consists of image patches corresponding to the bounding box of normal and abnormal objects extracted from the DB-I. Figure 1 shows a schematic diagram of the proposed iterative training process. We initially train our base detection model ($DM_0$) and binary classifier ($BC_0$) on the datasets DB-I and DB-O, respectively. The $DM_0$ is then deployed to a surveillance video stream for detecting abnormal events. The detected positive images are collected from the video stream and used for additional training to improve the performance of the current detection model. During each iteration of the iterative training process, the image dataset (DB-I) and object dataset (DB-O) are updated by incorporating additional training data from the video stream after the labels are corrected by the binary classifier. The detection model and binary classifier are iteratively trained on the updated datasets to become more adaptive to the new environment. The iterative learning process is repeated for $N$ iterations until the performance of the detection model reaches the desired level for the new surveillance video stream.

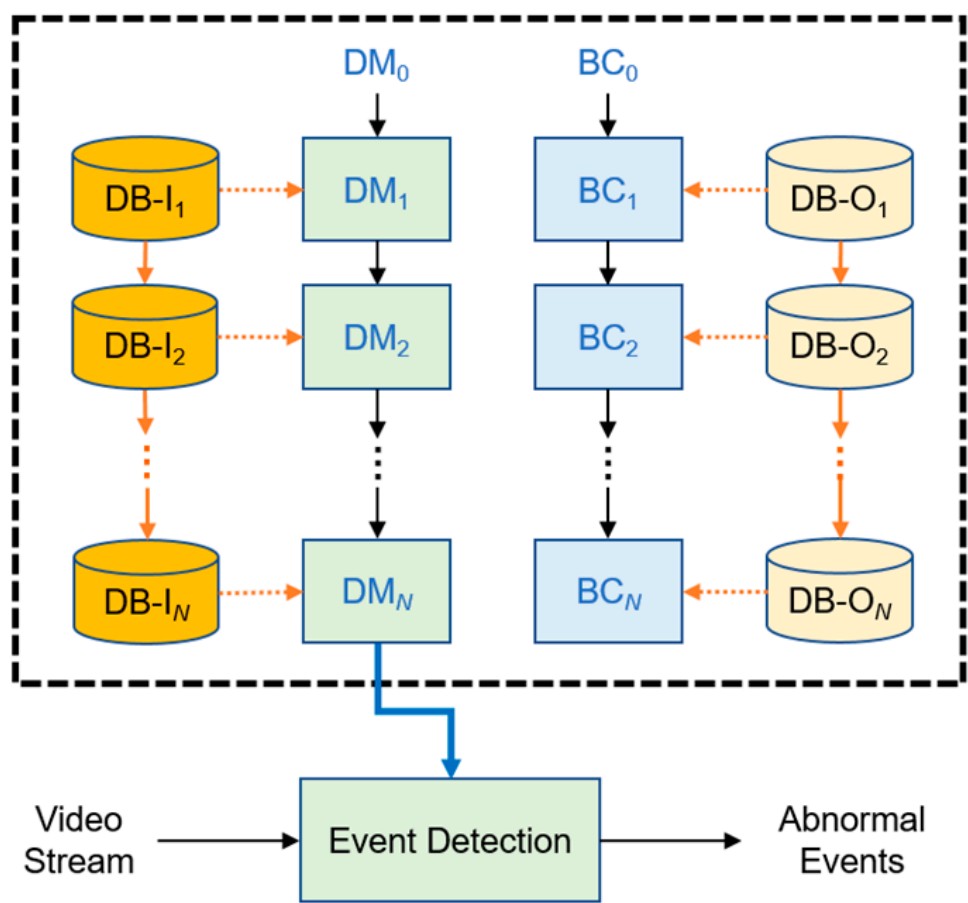

**Figure 1.** A schematic diagram of the proposed iterative training process.

Figure 2 shows the sequence of iterative learning with a binary classifier for an iteration. In the initial step of the iterative training, a detection model is trained with an initial training dataset (DB-I) to create our base detection model ($DM_0$). Additionally, a base binary classifier ($BC_0$) is trained using the initial training dataset (DB-O), which contains both normal and abnormal objects extracted from the DB-I. In the k-th iteration, given an image dataset ($DB-I_k$), the detection model from the previous iteration ($DM_{(k-1)}$) is trained with the $DB-I_k$ to obtain the $DM_k$. The trained detection model ($DM_k$), is then deployed to monitor the surveillance video stream for a certain duration.

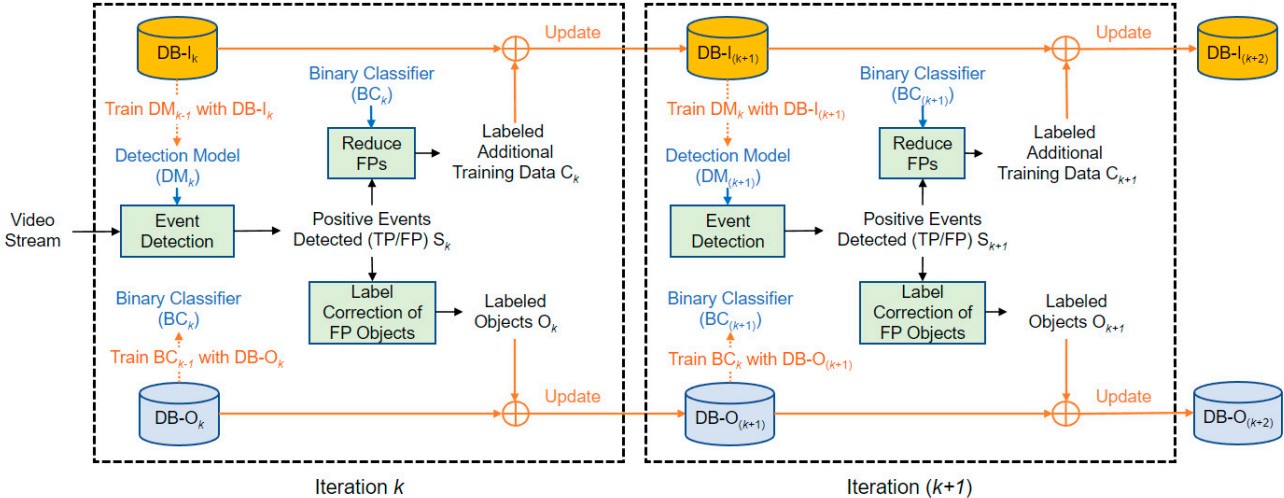

**Figure 2.** An iteration of the proposed iterative training with binary classifier.

During streaming, the detection model detects and stores all the abnormal events, denoted as $S_k$. Those positive images detected by the model contain true positives and false positives. Before using the positive images for additional training in the next iterative learning cycle, it is essential to separate the false positives and correct their labels. This ensures that the detection model can better learn the abnormal events in the next training cycle. To avoid the need for manual separation and the label correction of false positive images, the proposed iterative learning scheme utilizes a binary classifier. This binary classifier is trained to automatically detect false positives. All the images collected from the new video stream, after the labels are corrected, are added to the dataset ($DB-I_k$) to produce an updated image dataset ($DB-I_{(k+1)}$). In the k-th iteration, the binary classifier ($BC_{(k-1)}$) is trained with an object dataset ($DB-O_k$) to produce the $BC_k$. The binary classifier ($BC_k$) examines each object in the positive image to identify the false positive objects in the scene. The binary classifier corrects the labels of false positives in the additional streaming data and incorporates them into the training dataset ($DB-I_k$) to generate an updated training dataset ($DB-I_{k+1}$). This updated dataset is then used to train the detection model ($DM_k$). The datasets are updated iteratively as:

$$DB-I_{k+1} = DB-I_k \cup \left\{ C_i^{(k)} | \text{Labeled additional training image at iteration } k \right\} \quad (1)$$

$$DB-O_{k+1} = DB-O_k \cup \left\{ O_i^{(k)} | \text{Labeled additional object at iteration } k \right\} \quad (2)$$

The iterative training process for both the detection model and the binary classifier is repeated until the desired detection accuracy is achieved. If the binary classifier detects any false positives, it automatically changes the label of the abnormal object in that image to normal. Otherwise, the binary classifier keeps the label of all true positive events. For a set of positive images detected by the model in the *k*-th iteration, the binary classifier ($BC_k$) makes a decision for each object in the positive image, labeling them as either a true positive (TP) or a false positive (FP). The labels of the FP objects are corrected as negative.

The following formulas show the training steps for the detection model and the binary classifier in the next iteration ($k + 1$):

$$DM_{k+1} \leftarrow \text{Train } DM_k \text{ with DB-I}_{k+1} \tag{3}$$

$$BC_{k+1} \leftarrow \text{Train } BC_k \text{ with DB-O}_{k+1} \tag{4}$$

The binary classifier plays a crucial role in correcting the detection model's mistakes by classifying detected abnormal objects as either true positives or false positives. Figure 3 shows an image of a detected event containing an anomalous object labeled with a red bounding box and two non-anomalous objects labeled with green bounding boxes. Instead of manually labeling false positives to collect additional training data, the binary classifier automatically removes them from a set of additional training data without human intervention. The binary classifier ($BC_k$) then examines only the anomalous object and decides if the object is classified as true (anomalous) or false (non-anomalous). If the object is classified as true (anomalous), then the object is accepted as TP. If classified as false (non-anomalous), the object is considered an FP, and the detection label is corrected accordingly. All the positive images, along with their corrected object labels, are used to create a training dataset (DB-I$_{(k+1)}$) for the next training cycle. To update the dataset (DB-O$_k$) for training the binary classifier in the next iteration ($k + 1$), each true positive object is labeled as 1 and each false negative object as 0 from the detection event ($S_k$). All newly labeled objects ($O_k$) are added to the dataset (DB-O$_k$) to produce DB-O$_{(k+1)}$. Since the annotation process requires only two labels, the potential for errors and the need for human effort are reduced.

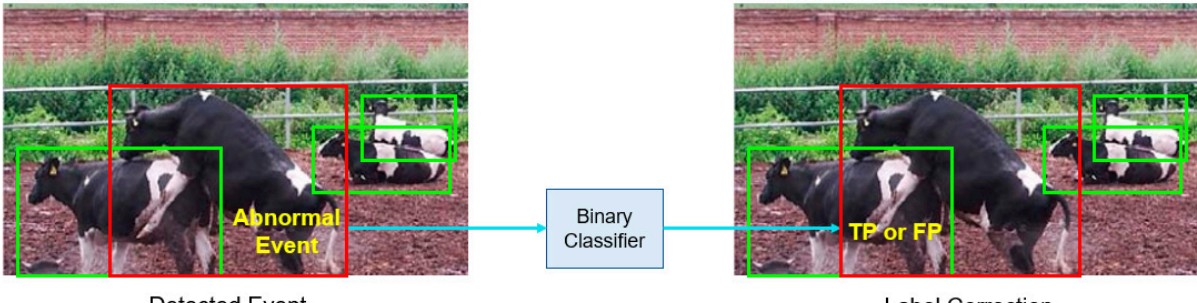

**Figure 3.** Correcting the label of a false positive object (red) detected using a binary classifier.

*3.2. The Model Architectures*

3.2.1. Detection Model

YOLOv5 [3] is a popular object detection model known for its fast and accurate capabilities in detecting objects. It utilizes convolutional neural networks (CNNs) to automatically extract features from images at different scales and positions, making it well-suited for object detection and recognition tasks. YOLOv5 has demonstrated high effectiveness and robustness in object detection, being able to process images in real-time and achieving a strong performance on various object detection benchmarks like the COCO Dataset. For our study, we used a nano model of YOLOv5 (YOLOv5n) as our base model to detect anomalous events. The flexibility of YOLOv5 allowed us to train it on custom datasets, enabling the detection of specific types of anomalies in particular contexts. Since the COCO Dataset might not include events of interest in specific applications, it became necessary to retrain the original YOLOv5 model with a set of training images containing abnormal events detected from a surveillance video stream. The resulting model served as a base detection model for detecting abnormal events. During the initial training, the model's weights were initialized to detect abnormal events of interest, and it was trained on two categories of normal and abnormal objects. The base YOLOv5 model was then deployed to a surveillance camera to collect streaming images and output the probabilities, objectness scores, and bounding boxes for each object on the screen. Anomalous events,

including both TPs and FPs, were corrected by the binary classifier to update the base dataset for the next iteration in the iterative training process.

3.2.2. Binary Classifier

In this paper, we utilized a pre-trained ResNet-18 model [23] as the binary classifier, which was trained on the ImageNet Dataset to classify 1000 object categories. The ResNet backbone enhances the model's performance by leveraging its capability to learn features from images effectively. The ResNet backbone can extract rich features from images and pass them to the final layers of the network, enabling accurate predictions. To adapt the ResNet model to our binary classification problem, the last output layer was replaced with a one-unit layer. This new output layer of the binary classifier was initialized with random weights, and during the initial epochs, only this layer was trained while keeping the weights in the backbone layers fixed. As the training progressed, the entire network was unfrozen, and training was carried out gradually. The binary classifier produced an abnormal probability score for the input object. To scale this output value to a range between 0 and 1, the sigmoid function was applied. Additionally, a threshold (defaulting to 0.5) was defined to determine whether the input object was a false positive. If the output value was greater than or equal to this threshold, the input object was considered abnormal. During the iterative training process, the binary classifier is crucial in distinguishing FPs from TPs, ensuring the accuracy of the detection model by correcting the labels and improving its performance.

## 4. Experimental Results

### 4.1. Data Collection

The effectiveness of the proposed iterative learning framework was validated through experiments focused on detecting the estrus behaviors of cows as an abnormal event from a surveillance video stream. To continuously monitor the cows for estrus behaviors, surveillance cameras were installed in eight cow barns, capturing video streams 24/7 during both daytime and nighttime. Each pen, approximately $8 \times 8$ m$^2$ in size, housed six cows with an average weight of 500 kg each. The installed surveillance cameras recorded video streams at a resolution of $1920 \times 1080$ at 60 frames per second (fps). Positioned at the corner of the cattle barn, the camera was placed 3 m above the ground level. Depending on the ambient light around the cow enclosure, RGB video data were recorded if the light exceeded a predefined value; otherwise, an infrared video was saved. In total, approximately 4392 h of video streaming data per camera were recorded over a period of 183 days, resulting in a total of 35,136 h of video data, captured during both daytime and nighttime. The streaming video was transferred to a Jetson Nano board through the Real-Time Streaming Protocol (RTSP) to provide training images for the base detection model.

Image frames were extracted from the video streams, and each cow object was then labeled as either "estrus" or "non-estrus" based on their behavior in the extracted image frame using the CVAT annotation tool [24]. This process was carried out to build a training dataset. The dataset was further divided into two separate sets to create two base detection models—one for daytime and the other for nighttime scenarios. Throughout the 183 days of the study, a total of 8431 images containing 113,209 instances of cows were collected from eight different video streams. From this large dataset of cow images, 6744 images were used for the iterative training of the base detection model, while 1687 images were reserved for evaluation purposes, following an 80:20 ratio. The distribution of the training and testing images for the two distinct types (daytime and nighttime) is presented in Table 2. The binary classifier was trained using cow objects extracted from the base training dataset, which comprised the 8431 cow images, as well as the FP images detected during the streaming. Table 3 provides the numbers of estrus cow objects for training and testing the binary classifier.

**Table 2.** Number of image samples in the datasets for the training and testing the detection model (DB-I).

|  | Training | | Testing | |
| --- | --- | --- | --- | --- |
|  | **Estrus** | **Non-Estrus** | **Estrus** | **Non-Estrus** |
| Daytime | 4503 | 844 | 1129 | 208 |
| Nighttime | 623 | 774 | 168 | 182 |
| Total | 5126 | 1618 | 1297 | 390 |

**Table 3.** Number of objects in the datasets for training and testing for the binary classifier (DB-O).

|  | Training | | Testing | |
| --- | --- | --- | --- | --- |
|  | **Estrus** | **Non-Estrus** | **Estrus** | **Non-Estrus** |
| Daytime | 4500 | 2064 | 1126 | 517 |
| Nighttime | 632 | 1038 | 158 | 260 |
| Total | 5132 | 3102 | 1284 | 777 |

*4.2. Building a Base Model for Estrus Cow Detection*

4.2.1. Detection Model Implementation

The architecture of our base model, YOLOv5n, designed for detecting estrus cows, was implemented using the PyTorch framework [25]. We used pre-trained model weights for YOLOv5n downloaded from the official GitHub repository [3]. These weights were trained on the COCO Dataset and served as a starting point for our object detection task using the transfer learning technique. Our experiments were conducted on a system running Ubuntu 18.04.5 LTS x64, with CUDA 11.6 and OpenCV 4.5.5. The hardware configuration included an Intel(R) Xeon(R) E5-2620 v4 @ 2.10 GHz processor with 128 GB of RAM. To accelerate the training process for YOLOv5n, we used four NVIDIA TITAN Xp GPUs, each with 12 GB of memory. For the task of streaming video and detecting estrus images, a Jetson Nano Developer Kit was employed, which features a Quad-core ARM A57 CPU @ 1.43 GHz, 4 GB of RAM, and a 128-Core Maxwell GPU. In our training setup, the learning rate was set to 0.01 with an SGD optimizer. A decay value of 0.001 was used to prevent overfitting. The binary classifier was trained for 50 epochs, while YOLOv5n was trained for 100 epochs. The object confidence threshold was set to 0.5, and the IoU (Intersection over Union) threshold for non-maximum suppression (NMS) was set to 0.45. These parameters were chosen to optimize the detection performance of our model.

Due to the distinct monitoring environments during the daytime and nighttime, two identical base detection models were employed—one for daytime monitoring and the other for nighttime monitoring. This approach allowed us to adapt the models specifically to the characteristics of each scenario. The daytime model was used to analyze the RGB streaming video containing color information, which can be useful for identifying abnormal objects and tracking changes in the environment. On the other hand, the nighttime model was used on the infrared video stream, enabling the detection of object contours and shapes. For consistent labeling during inference, the detection model decides on a positive estrus event based on the trailing moving average of the detection results for multiple image frames. A non-estrus image contains no estrus cows in the frame. In a sequence of image frames and their corresponding detection labels (positive or negative), the detection model determines the label of each frame based on the majority voting of the detection labels of the past few video frames. To achieve accurate detection of an estrus event, which usually lasts from 2 to 3 s, a moving average calculation is utilized over a sequence of 10 consecutive video frames. This method enhances the precision of the detection process by considering temporal information.

4.2.2. Training the Base Detection Models

The base detection model, $DM_0$, was trained using the transfer learning technique and the pre-trained weights of the COCO Dataset. The training process took approximately 5 h to complete and resulted in a mean average precision (mAP@0.5) of 0.98 and 0.968 for the daytime model and nighttime model, respectively. These results indicate that the detection models performed effectively on the base estrus dataset. Table 4 shows the training results for both models. Each surveillance camera was monitoring a specific cow enclosure with unique environmental conditions. Separate daytime and nighttime models were trained for each video stream. Both detection models participated in the iterative training process using the initial weights from their base models. The focus was on evaluating the accuracy of the models in detecting estrus images and observing the performance improvements in each iteration cycle. The image-level accuracy of the base models on the test set was calculated using the base training dataset. Table 5 shows the accuracy measurement for both the daytime and nighttime models at different object confidence thresholds. The daytime model has higher accuracy than the nighttime model due to the large number of RGB training images used during the fine-tuning process.

**Table 4.** Training results of base detection model ($DM_0$).

|  | **Precision** | **Recall** | **mAP@0.5** | **mAP@0.5:0.95** |
|---|---|---|---|---|
| Daytime | 0.968 | 0.967 | 0.984 | 0.794 |
| Nighttime | 0.945 | 0.93 | 0.968 | 0.758 |

**Table 5.** Accuracy of base detection model ($DM_0$) at different confidence thresholds.

|  | **Confidence Thresholds** | | | |
|---|---|---|---|---|
|  | **0.25** | **0.5** | **0.75** | **0.8** |
| Daytime | 0.991 | 0.990 | 0.938 | 0.901 |
| Nighttime | 0.977 | 0.974 | 0.929 | 0.891 |

Two separate binary classifiers were trained for daytime and nighttime. The classifiers were fine-tuned using the ResNet-18 backbone, incorporating pre-trained weights sourced from the ImageNet Dataset. The training process took approximately 2 h to complete. Each classifier was assessed on the testing set by applying a threshold value of 0.5 to determine the classification results. If the network's output value was greater than or equal to the threshold, the input object was classified as an estrus; otherwise, it was classified as a non-estrus. The classification accuracy was 98.6% for the daytime model and 96.3% for the nighttime model. The nighttime model exhibited signs of overfitting, which could be attributed to the relatively smaller size of its training data compared to the daytime model.

*4.3. Iterative Training of the Detection Model*

Throughout the six weeks of video streaming, the iterative training process was employed to monitor two testing surveillance cameras. However, there was a significant drop in accuracy during the first cycle of iterative training. When deployed in unfamiliar environments, such as the two new testing video streams, the accuracy decreased rapidly, and the base models produced a large number of false positives in the first week. These false positives were then used as inputs for the base binary classifiers. The daytime binary classifier corrected the labels for the RGB image data, while the nighttime binary classifier corrected the labels for the IR image data. An increase in accuracy was observed during the next five weeks of video streaming. Figure 4 shows a visual example of the detection results when applying the iterative training algorithm to reduce false positive detections.

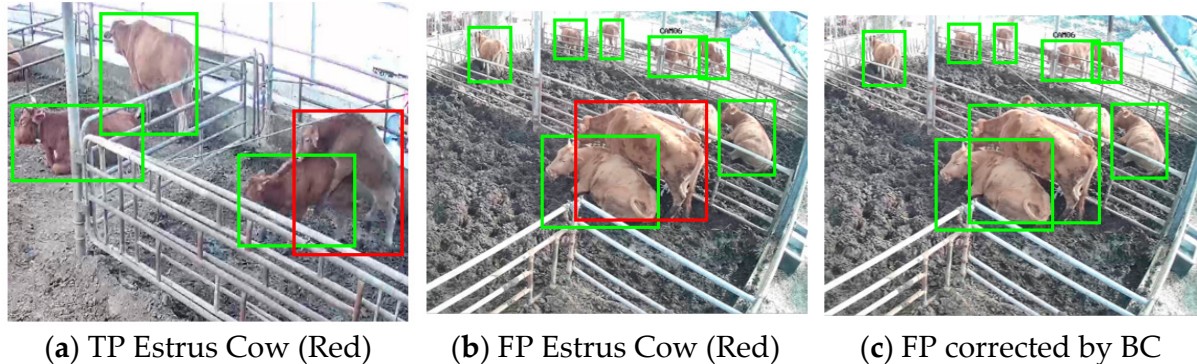

(**a**) TP Estrus Cow (Red)     (**b**) FP Estrus Cow (Red)     (**c**) FP corrected by BC

**Figure 4.** False positive correction during the iterative training process. (**a**) The detection model correctly identifies a true positive estrus cow (red) along with normal cow objects (green); (**b**,**c**) The model incorrectly identifies a cow standing closely as an estrus object, resulting in a false positive (**b**); the FP is later corrected by the binary classifier (BC) during the iterative training process (**c**).

Figure 5 illustrates the accuracy performance for two testing video streams over five cycles (five weeks) of iterative training. In test video stream 1, the base models saw a significant decrease in accuracy, dropping from 99.0% and 97.4% to 35.7% and 0.0% for the daytime and nighttime models, respectively. However, after the first cycle of training, the accuracy of the daytime model ($DM_1$) made a significant improvement, jumping from 35.7% to 82.4%. The nighttime model ($DM_1$) also saw an increase in accuracy, going from 0.0% to 65.3%. As the iterative training process continued, the accuracy of both models increased rapidly. The nighttime model ($DM_2$) was the first to reach 100% accuracy, followed by the daytime model ($DM_5$), which eventually also achieved 100% accuracy. In test video stream 2, the base models experienced a similar decline in accuracy as in the first stream. The accuracy of the daytime model ($DM_0$) decreased from 99.0% to 68.6%, while the accuracy of the nighttime model ($DM_0$) dropped from 97.4% to 30.5%. However, the daytime model quickly adapted to the new environment during the iterative training process and reached 100% accuracy at $DM_2$. It maintained this accuracy for the next three weeks. In contrast, the accuracy of the nighttime model increased gradually between $DM_1$ and $DM_4$ and eventually reached 100% accuracy at $DM_5$.

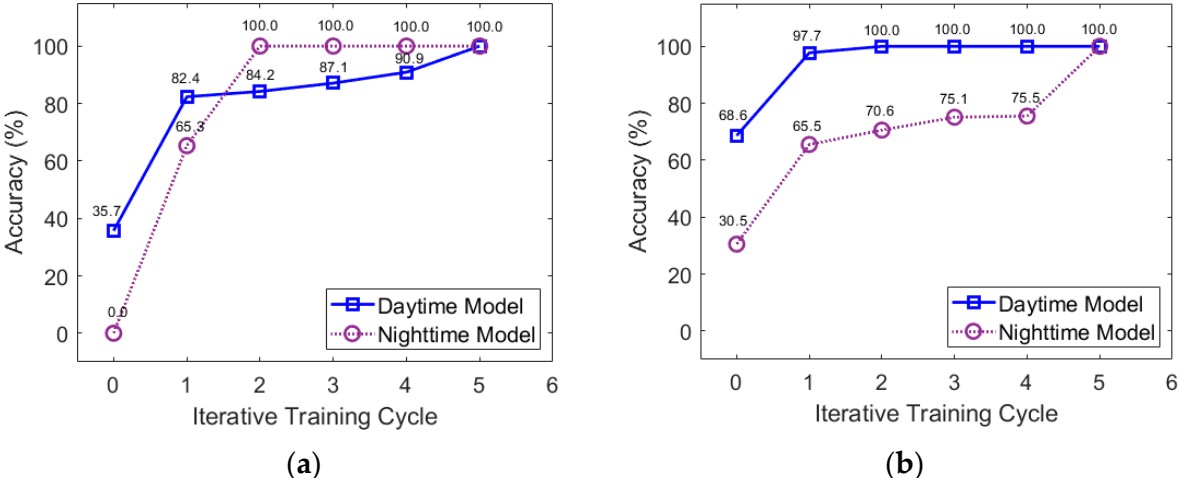

(**a**)     (**b**)

**Figure 5.** Estrus cow detection accuracies during the iterative training cycles for different video streams. (**a**) Test stream 1: Both models' accuracy rapidly increases as iterative training cycles are repeated. (**b**) Test stream 2: The daytime model promptly reached its highest accuracy, while the nighttime model achieved its highest accuracy more gradually.

A test was conducted to validate the accuracy of our binary classifiers throughout the training process. These classifiers were divided into two categories: daytime and nighttime. The classifiers were used to correct the estrus labels in the detected images in two testing streams. Figure 6 illustrates the accuracy of both the daytime and nighttime classifier models in correcting the estrus labels. The accuracy remained stable at around 98% and 96% for the daytime and nighttime classifier models, respectively. Table 6 shows the performance metrics of the two binary classifiers in different iteration steps. The precision, recall, F1-score, and accuracy metrics exceeded 90% for both classifiers. Notably, the daytime model outperformed the nighttime model. These values suggest that our binary classifiers were unaffected by the environment and could be trained independently of the detection models.

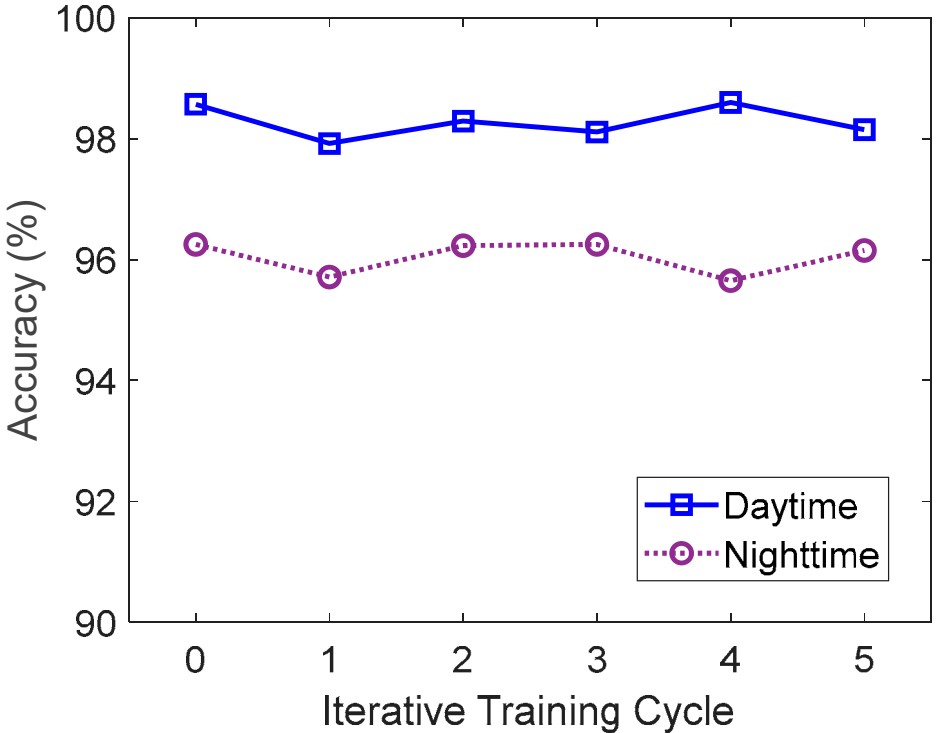

**Figure 6.** Accuracy of the binary classifiers.

**Table 6.** Performance metrics of the binary classifiers at each iterative training cycle.

| Models | Metrics | Iteration ($k$) | | | | | |
|---|---|---|---|---|---|---|---|
| | | 0 | 1 | 2 | 3 | 4 | 5 |
| Daytime | Precision | 0.975 | 0.966 | 0.974 | 0.967 | 0.971 | 0.968 |
| | Recall | 0.985 | 0.976 | 0.977 | 0.979 | 0.989 | 0.980 |
| | F1-score | 0.980 | 0.971 | 0.976 | 0.973 | 0.979 | 0.974 |
| | Accuracy | 0.986 | 0.979 | 0.983 | 0.981 | 0.986 | 0.982 |
| Nighttime | Precision | 0.973 | 0.966 | 0.964 | 0.964 | 0.958 | 0.971 |
| | Recall | 0.977 | 0.976 | 0.987 | 0.987 | 0.977 | 0.976 |
| | F1-score | 0.975 | 0.971 | 0.975 | 0.975 | 0.967 | 0.973 |
| | Accuracy | 0.962 | 0.955 | 0.962 | 0.963 | 0.950 | 0.962 |

The accuracy of our detection model in identifying estrus cows was assessed using the mean average precision (mAP@0.5). This metric provides a comprehensive evaluation of the model's performance in detecting the desired events. Table 7 shows the mAP@0.5 at different stages of the iterative training process for both testing video streams. The mAP@0.5 initially decreased in the first cycles for both testing streams. However, during

the training process, the mAP gradually increased and peaked at 99.5%. The trend of increasing mAP was consistent with the increasing accuracy in estrus detection. The human effort required to correct the labels of estrus cow images are also estimated. Instead of using the binary classifier, manual labeling corrects all the estrus labels from the streaming data by using human annotation. Three human operators were employed to manually verify each abnormal event detected by the object detector and label them as TP or FP. During the manual correction, we timed the annotation process required to verify all instance objects in abnormal images and obtained an average time from all experts as a final estimation time in seconds. Table 8 shows the comparison between the time needed for manual annotation and the time needed for annotation using the binary classifier at different stages of the training process. As the number of estrus events varies from week to week, the number of collected streaming images also varies. The annotation time for both the daytime and nighttime data was calculated. The binary classifier proved to be much more efficient than human operators in completing the annotation process.

**Table 7.** Mean average precision (mAP@0.5) of estrus cow detection at each iterative training cycle.

| Test Stream | Detection Models | Iteration ($k$) | | | | | |
|---|---|---|---|---|---|---|---|
| | | 0 | 1 | 2 | 3 | 4 | 5 |
| Stream 1 | Daytime | 0.737 | 0.964 | 0.988 | 0.989 | 0.991 | 0.995 |
| | Nighttime | 0.00 | 0.955 | 0.995 | 0.995 | 0.995 | 0.995 |
| Stream 2 | Daytime | 0.951 | 0.985 | 0.995 | 0.995 | 0.995 | 0.995 |
| | Nighttime | 0.761 | 0.955 | 0.959 | 0.963 | 0.963 | 0.995 |

**Table 8.** Comparison of annotation time of the binary classifier and manual annotation (s).

| Test Stream | Iteration ($k$) | Number of Additional Training Images | Manual Labeling | | | | Binary Classifier |
|---|---|---|---|---|---|---|---|
| | | | Expert 1 | Expert 2 | Expert 3 | Average | |
| Stream 1 | 0 | 184 | 2036 | 2025 | 2040 | 2034 | 12 |
| | 1 | 48 | 481 | 451 | 460 | 464 | 5 |
| | 2 | 56 | 601 | 573 | 581 | 585 | 5 |
| | 3 | 43 | 344 | 341 | 337 | 341 | 4 |
| | 4 | 32 | 298 | 300 | 295 | 895 | 3 |
| | 5 | 15 | 152 | 148 | 157 | 152 | 2 |
| Stream 2 | 0 | 110 | 906 | 912 | 910 | 909 | 10 |
| | 1 | 190 | 1395 | 1382 | 1401 | 1393 | 13 |
| | 2 | 20 | 188 | 185 | 192 | 188 | 3 |
| | 3 | 161 | 1264 | 1257 | 1280 | 1267 | 11 |
| | 4 | 79 | 269 | 256 | 271 | 265 | 8 |
| | 5 | 75 | 265 | 251 | 268 | 261 | 7 |

A paired t-test was conducted to determine whether there was a significant difference in the performance of the detection model when trained using the iterative training with a binary classifier and non-iterative training. The paired t-test aims to assess whether there is a significant discrepancy in the means of the evaluation metrics between the two training methods. The null hypothesis (H0) posits that there would be no significant difference in the mean accuracy of the model when trained with both training schemes. The alternative hypothesis (Ha) suggests that there would be a substantial difference in the mean accuracy of the model when trained using the two learning schemes. For the daytime model applied to video stream 1, the average accuracies of iterative training over five iterations and the baseline were 80.05% and 35.7%, respectively, resulting in a t-value of 4.806. At a 95% confidence level, the calculated t-value falls outside the critical region of $-2.571, 2.571$. As a result, the null hypothesis is rejected, indicating a significant difference between the two training schemes. The same conclusion holds with a 99% confidence level.

## 5. Ablation Study

### 5.1. Iterative Training with Manual Labeling

This study aimed to evaluate the effectiveness of our iterative training approach, which incorporates a binary classifier, in comparison to a manual labeling process where humans directly corrected estrus images from the streaming data. Our proposed iterative training algorithm is designed to work with various object detectors, such as YOLOv5, SSD, or Faster-RCNN. For this study, we focused on YOLOv5, as it is well-suited for edge computing devices. To account for different lighting conditions, separate models were employed for daytime and nighttime video streams. Both models underwent an iterative training process consisting of five cycles ($k = 0, \ldots, 5$). Table 9 presents a comparison of the accuracy obtained by both training processes for the two video streams. Notably, the manual daytime model and nighttime model were initialized with the same weights as the corresponding models that utilized a binary classifier, as they were trained on the same base training dataset.

**Table 9.** Accuracy comparison between iterative training with a binary classifier and with manual labeling in two different testing video streams.

| (a) Test Stream 1 | | | | | | | |
|---|---|---|---|---|---|---|---|
| **Models** | **Types** | **Iteration ($k$)** | | | | | |
| | | **0** | **1** | **2** | **3** | **4** | **5** |
| Daytime | Binary Classifier | 0.357 | 0.824 | 0.842 | 0.871 | 0.909 | 1.0 |
| | Manual Labeling | 0.357 | 0.836 | 0.852 | 0.884 | 0.938 | 1.0 |
| Nighttime | Binary Classifier | 0.0 | 0.653 | 1.0 | 1.0 | 1.0 | 1.0 |
| | Manual Labeling | 0.0 | 0.753 | 1.0 | 1.0 | 1.0 | 1.0 |
| (b) Test Stream 2 | | | | | | | |
| **Models** | **Types** | **Iteration ($k$)** | | | | | |
| | | **0** | **1** | **2** | **3** | **4** | **5** |
| Daytime | Binary Classifier | 0.686 | 0.977 | 1.0 | 1.0 | 1.0 | 1.0 |
| | Manual Labeling | 0.686 | 0.981 | 1.0 | 1.0 | 1.0 | 1.0 |
| Nighttime | Binary Classifier | 0.305 | 0.655 | 0.706 | 0.751 | 0.755 | 1.0 |
| | Manual Labeling | 0.305 | 0.667 | 0.753 | 0.914 | 1.0 | 1.0 |

Overall, the accuracy of the manual detection models for both daytime and nighttime was slightly higher than the corresponding models using a binary classifier. Notably, the manual nighttime model achieved the highest accuracy in the early stages, such as the model $DM_4$ in test stream 2, which reached 100% accuracy, while the nighttime model using a binary classifier $DM_4$ achieved 75.5% accuracy. However, as discussed in Section 4, the manual detection model requires a significant amount of time for annotation, which becomes increasingly burdensome as the farm expands to encompass hundreds of cow enclosures. In other words, while the iterative training process with manual labeling may achieve high accuracy faster than the process using a binary classifier, it is not an efficient way to reduce the human effort required for labeling the streaming data, especially in a large-scale livestock farming operation.

### 5.2. Positive Training versus Negative Training

An experiment was conducted to compare the effects of positive training and negative training during a one-week video streaming period. The objective was to assess the impact of these training approaches on the performance of the model in detecting the desired events. Starting from the base model, we collected the streaming data, labeled them using the binary classifier, and started the fine-tuning process. The collected data were separated into two sets: (1) positive samples, which are images with at least one estrus cow appearing on the scene, and (2) negative samples, which contained only normal cows with no estrus

behaviors. With positive learning, the detection model was only trained on positive cow images while ignoring the negative ones. As for negative learning, we exclusively utilized negative cow images to train the detection model. After one week of streaming, the negative model was unable to detect any estrus cows. Despite inheriting weights from the base model, which was originally trained with positive images, the negative models were not able to detect estrus cows in the streaming video. This is because they adapted their weights to learn only negative cases and ignored all positive cases. In contrast, both the positive model and the combined model were able to detect estrus cows in the video. Figure 7 shows a comparison of the accuracy between the positive model and the combined model at different timestamps.

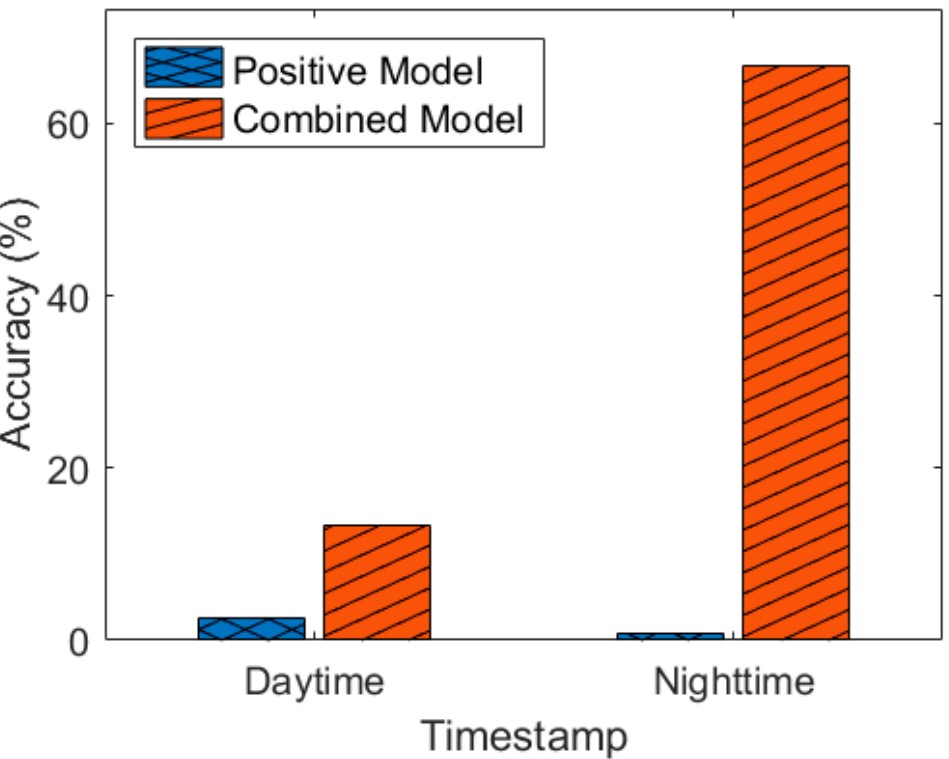

**Figure 7.** Accuracy comparison between the positive model and the combined models at different timestamps.

The positive model detected a higher number of false positives compared to the combined models, despite detecting the same number of true positive images. This disparity occurred because the positive model solely learned from the estrus samples and did not have the opportunity to learn from the negative samples. Consequently, its accuracy dropped to 2.55% and 0.68% for daytime and nighttime, respectively. In contrast, the combined model was able to learn from both positive and negative samples, resulting in an accuracy of 13.26% and 66.67% for the daytime and nighttime models, respectively. This demonstrates that the detection model trained on both positive and negative samples outperforms one trained solely on positive samples.

## 6. Conclusions

This study proposes an iterative training approach that leverages a binary classifier to improve the accuracy of object detection in identifying abnormal objects, specifically estrus cows. By utilizing the lightweight YOLOv5 object detection model on edge devices like the Jetson Nano, we successfully monitored surveillance videos during both daytime and nighttime conditions. The streaming data collected were labeled by a binary classifier, and both the detection models and the binary classifiers underwent an iterative training process

to adapt and fine-tune their weights according to changing environmental conditions. Our approach enabled the real-time monitoring and detection of abnormal events, such as estrus cows, in challenging surveillance scenarios. The use of a binary classifier significantly reduced the need for human intervention in the labeling process, thereby alleviating the workload associated with manually correcting incorrect labels (false positives) during training. Our demonstration showed that employing a binary classifier substantially decreases the time and effort required for correcting image labels, making it a more efficient alternative to manual labeling. Moreover, our approach achieved comparable accuracy and gradually reached the highest performance within a reasonable timeframe. The binary classifiers can also be trained independently of the detection models, making them reusable and upgradeable modules for large-scale surveillance videos. Our approach for estrus cow detection was validated using two test video streams, achieving impressive results with an estrus accuracy of up to 100% and a mAP@0.5 of 0.995. Additionally, the annotation time was reduced by up to 98% compared to manual labeling. For future work, our plan includes enhancing the YOLOv5 models by incorporating other real-time object detectors and testing them on various edge devices. This expansion aims to assess the effectiveness of our proposed iterative training in different setup environments.

**Author Contributions:** Conceptualization, S.G.K.; Methodology, C.H.T.; Validation, C.H.T.; Investigation, C.H.T.; Writing—original draft, C.H.T.; Writing—review & editing, S.G.K.; Supervision, S.G.K.; Funding acquisition, S.G.K. All authors have read and agreed to the published version of the manuscript.

**Funding:** This research was supported by the Institute of Information and Communications Technology Planning and Evaluation (IITP) grant funded by MSIT of Korea under Grant 2019–0-00231 (Development of Artificial Intelligence Based Video Security Technology and Systems for Public Infrastructure Safety).

**Conflicts of Interest:** The authors declare no conflict of interest.

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
