# Peer review of "An Iterative Learning Scheme with Binary Classifier for Improved Event Detection in Surveillance Video"

_electronics, doi:10.3390/electronics12153275_

Round 1
Reviewer 1 Report
The proposed manuscript presents an iterative approach for training a deep learning model for abnormal behavior detection in surveillance videos. A binary classifier is used to automate the labeling of true positives and false positives. Although the approach is not fundamentally new, it is very interesting to see its application to model retraining and surveillance video analysis.
What limits the interest is the application to the detection of estrus behavior in cows. First, this is an event that is not of interest to the majority of the population. It may not even be an abnormal behavior for cattle breeders. As far as I know, breeding takes place under very controlled conditions and is not something that should be recognized as an unexpected event. If it is not, the paper should explain why it is important to recognize these events (1). Second, the event is very easy to detect because the contour of the detected object changes significantly. It would be much more interesting to test the proposed model on some human activities that can be considered abnormal and that would help to increase the security of some detection systems.
The manuscript should highlight how long it takes to train the model and classifier after each iteration, and why a period of one week was chosen to collect data for the next iteration of training (2). In L358, it is claimed that the initial training took about 5 hours for the model and 2 hours for two separate classifiers (for day and night). It would be interesting to show the entire system allowing 24-hour monitoring with automatic retraining and model renewal without stopping the system. Of course, this is beyond the scope of this manuscript. It would also be interesting to determine the optimal retraining period to accelerate convergence.
YOLOv5 was used for object recognition. It would be good to emphasize why this particular version is used (3), considering that the current version is 8.
Reference 3 contains a number of errors (4). It should be corrected.
The English is generally good. Minor corrections are needed, however, as some sentences lack the plural, articles, and conjunctions.
Author Response
Comment 1: As far as I know, breeding takes place under very controlled conditions and is not something that should be recognized as an unexpected event. If it is not, the paper should explain why it is important to recognize these events.
Response: Recognizing estrus events of cows through video surveillance monitoring is of utmost importance for livestock farmers. While breeding can be managed under controlled conditions, accurately tracking estrus behaviors of cows is essential to determine the optimal timing for breeding. Estrus events occur infrequently, making continuous monitoring with surveillance cameras indispensable to identify the most suitable period for successful breeding. In the Introduction, we have provided a comprehensive explanation of the significance of recognizing estrus events through surveillance monitoring, specifically from lines 37 to 40.
Comment 2: The manuscript should highlight how long it takes to train the model and classifier after each iteration, and why a period of one week was chosen to collect data for the next iteration of training.
Response: Training the model and the classifier takes a few hours at each iteration; however, we need to collect a certain number of estrus images for effective model training. In our experiment settings, we had to wait for a week to accumulate 15-100 estrus images. Considering that we repeat several iterative training cycles up to 5 times, we aimed to complete the training within a one-month timeframe.
Comment 3: YOLOv5 was used for object recognition. It would be good to emphasize why this particular version is used, considering that the current version is 8.
Response: In this study, our primary goal was to develop a lightweight detection model that could be effectively implemented on edge computing devices, such as Jetson Nano. For this purpose, we chose the nano model of YOLOv5 (YOLOv5n), known for its lightweight characteristics and effortless implementation on Jetson Nano. This makes it an ideal and cost-effective solution for cow barn monitoring.
Comment 4: Reference 3 contains a number of errors. It should be corrected.
Response: We fixed the errors in Reference 3.
Comment 5: The English is generally good. Minor corrections are needed, however, as some sentences lack the plural, articles, and conjunctions.
Response: We proofread the manuscript and fixed grammar errors.
Reviewer 2 Report
Dear authors:
Thank you for submitting your article titled "An Iterative Learning with Binary Classifier for Improved Event Detection in Surveillance Video" for review.
The paper is well organized and fits within the journal's scope. I have some minor comments for improvement of the manuscript.
1. It would be beneficial to include visual examples of the detection results to demonstrate how the proposed iterative algorithm effectively reduces false positive detections.
2. Please enhance Figure 4 by adding numerical labels to the key points for better clarity and easier understanding.。
3. Please provide additional clarification on the definition and calculation of the "manual correction time" mentioned in line 437.
4. Consider conducting ablation experiments by replacing the YOLO5 backbone network with other backbone networks to evaluate the generalizability of the proposed iterative strategy.
5. Please include future work directions in the conclusion to provide valuable guidance for further advancements in this field.
I hope that my comment is useful to improve the quality of this manuscript.
Best regards
Author Response
Comment 1: It would be beneficial to include visual examples of the detection results to demonstrate how the proposed iterative algorithm effectively reduces false positive detections.
Response: We have included visual examples in a Figure 4 to demonstrate the detection results of reducing false positives with the proposed iterative training approach.
Comment 2: Please enhance Figure 4 by adding numerical labels to the key points for better clarity and easier understanding.
Response: We added numerical labels to the graphs in revised Figure 5, as suggested.
Comment 3: Please provide additional clarification on the definition and calculation of the "manual correction time" mentioned in line 437.
Response: Manual correction time refers to the time taken by a human operator to assign a correct label to an object detected by the model with an incorrect label. We measured and averaged the time in seconds for three human operators to label all false positive objects in the detected images for two test video streams. Table 8 has been included to provide a comprehensive breakdown of the annotation times for manual correction and the binary classifier, offering detailed insights into the process.
Comment 4: Consider conducting ablation experiments by replacing the YOLO5 backbone network with other backbone networks to evaluate the generalizability of the proposed iterative strategy.
Response: Our proposed iterative training is compatible with various object detectors, including YOLO, SSD, and Faster-RCNN. For the current study, we validated the effectiveness of our scheme using YOLOv5, as it can be implemented on edge computing devices. In the ablation study 1 – line 483, we indicated that other backbone networks can be used to replace YOLOv5 for enhanced generalizability.
Comment 5: Please include future work directions in the conclusion to provide valuable guidance for further advancements in this field.
Response: We have included future work directions in the conclusion of the revised manuscript. Our plan is to test the proposed iterative training on other real-time object detectors that can be implemented on various edge devices.
Reviewer 3 Report
In this study, the authors have presents an iterative training framework with a binary classifier to improve the learning capability of a deep learning model for detecting abnormal behaviors in surveillance video, It is interesting. To ensure the desired performance in new environments, this paper proposes a binary classifier which is trained on bounding boxes extracted from the detection model, and automatically label false positive data without human intervention. In addition, there are some irregularities in writing and errors in presentation in the manuscript, and the authors are advised to carefully review and revise it. Although there are some problems, publication is still considered.
1. Line 209: Missing 'should be' between 'labels' and' corrected '.
2. Line 270: "including both true positives and false positives, are corrected" should be "including both true positives and false positives, is corrected" to match the singular subject "Anomalous events."
3. Line 338: "is 50, 100" should be "are 50 and 100" for subject-verb agreement.
4. Line 348: "smoothen" should be "smooth" for proper grammar.
5. Line 347: "confident thresholds" should be "confidence thresholds" for proper grammar.
6. Line 308: "real-time streaming protocol (RTSP)" should be "Real-Time Streaming Protocol (RTSP)" with initial caps for the proper noun.
7. Line 329: "COCO dataset" should be "COCO Dataset" with initial caps for the proper noun.
8. Line 333: "Nvidia TITAN Xp GPUs" should be "NVIDIA TITAN Xp GPUs" with initial caps for the proper noun.
9. Line 336: "128-core Maxwell GPU" should be "128-Core Maxwell GPU" with initial caps for the proper noun.
10. Line 375: "ImageNet dataset" should be "ImageNet Dataset" with initial caps for the proper noun.
11. Line 274: "desired accuracy on the streaming video" should be "desired accuracy for the streaming video" to clarify that the accuracy is being measured for the video stream, not achieved on the video itself.
12. Line 306: "There is a total of 35,136 hours of video data were obtained" should be "A total of 35,136 hours of video data was obtained" to match the singular subject "total."
Author Response
Comment 1: Line 209: Missing 'should be' between 'labels' and' corrected '.
Response: We added the missing words as suggested.
Comment 2: Line 270: "including both true positives and false positives, are corrected" should be "including both true positives and false positives, is corrected" to match the singular subject "Anomalous events."
Response: We fixed it as suggested.
Comment 3: Line 338: "is 50, 100" should be "are 50 and 100" for subject-verb agreement.
Response: We fixed it as suggested.
Comment 4: Line 348: "smoothen" should be "smooth" for proper grammar.
Response: We changed the sentence to “For consistent labelling during the inference,” for clarity.
Comment 5: Line 347: "confident thresholds" should be "confidence thresholds" for proper grammar.
Response: We fixed it as suggested.
Comment 6: Line 308: "real-time streaming protocol (RTSP)" should be "Real-Time Streaming Protocol (RTSP)" with initial caps for the proper noun.
Response: We fixed it as suggested.
Comment 7: Line 329: "COCO dataset" should be "COCO Dataset" with initial caps for the proper noun.
Response: We fixed it as suggested.
Comment 8: Line 333: "Nvidia TITAN Xp GPUs" should be "NVIDIA TITAN Xp GPUs" with initial caps for the proper noun.
Response: We capitalized the words as suggested.
Comment 9: Line 336: "128-core Maxwell GPU" should be "128-Core Maxwell GPU" with initial caps for the proper noun.
Response: We capitalized the word as suggested.
Comment 10: Line 375: "ImageNet dataset" should be "ImageNet Dataset" with initial caps for the proper noun.
Response: We fixed it as suggested.
Comment 11: Line 274: "desired accuracy on the streaming video" should be "desired accuracy for the streaming video" to clarify that the accuracy is being measured for the video stream, not achieved on the video itself.
Response: We fixed it as suggested.
Comment 12: Line 306: "There is a total of 35,136 hours of video data were obtained" should be "A total of 35,136 hours of video data was obtained" to match the singular subject "total."
Response: We fixed it as suggested.
Reviewer 4 Report
-I like that authors point out the contributions of their work. However, I suggest that in the related work section, authors summarize the difference with other approaches in a table, describing advantages and/or disadvantages with other approaches to related work.
-The authors compare Binary Classifier Manual on Daytime and Nighttime using an “accuracy” metric. However, to conclude robustly, the authors must include metrics such as recall, precision, accuracy, f-measure, since by only using accuracy, if in the training data there is a majority class than another, the classifier tends to predict that value of class if the classes are unbalanced. I ask the authors to include the results of these metrics in order to strengthen their conclusions.
-I suggest that the authors make the test datasets available, so that this work can be cited and the results can be replicated and improved.
-When the authors state in the conclusions “We demonstrated that the use of a binary classifier can significantly reduce the time and effort required to correct incorrect 517 image labels (false positives) during the training process”, there is no evidence that the authors have taken the time to quantify or demonstrate that there really was a reduction, If the authors timed the times, they should include it.
-Finally, regarding the previous point, I suggest the authors provide evidence that their proposal significantly improves the results, with some statistical test.
Author Response
Comment 1: I like that authors point out the contributions of their work. However, I suggest that in the related work section, authors summarize the difference with other approaches in a table, describing advantages and/or disadvantages with other approaches to related work.
Response: We have compared our proposed scheme with other approaches in Table 1, as suggested.
Comment 2: The authors compare Binary Classifier Manual on Daytime and Nighttime using an “accuracy” metric. However, to conclude robustly, the authors must include metrics such as recall, precision, accuracy, f-measure, since by only using accuracy, if in the training data there is a majority class than another, the classifier tends to predict that value of class if the classes are unbalanced. I ask the authors to include the results of these metrics in order to strengthen their conclusions.
Response: We have included performance metrics such as recall, precision, F1-score, and accuracy in Table 6, as suggested.
Comment 3: I suggest that the authors make the test datasets available, so that this work can be cited and the results can be replicated and improved.
Response: Thank you for your suggestion. We appreciate your interest in accessing the dataset used in our work. Unfortunately, we are not allowed to publish the dataset as per the request by our sponsor. We apologize for not being able to follow your suggestion in this regard. We hope you understand the constraints we are facing, and we value your understanding in this matter.
Comment 4: When the authors state in the conclusions “We demonstrated that the use of a binary classifier can significantly reduce the time and effort required to correct incorrect 517 image labels (false positives) during the training process”, there is no evidence that the authors have taken the time to quantify or demonstrate that there really was a reduction, If the authors timed the times, they should include it.
Response: In Table 8, we compared the annotation time between the binary classifier and manual annotation.
Comment 5: Finally, regarding the previous point, I suggest the authors provide evidence that their proposal significantly improves the results, with some statistical test.
Response: Our binary classifiers demonstrated their effectiveness in detecting estrus objects, as evidenced by achieving a mean average precision of 0.995 and performance metrics such as precision, recall, accuracy, and F1-score all surfacing 95%.
Reviewer 5 Report
The manuscript proposes an iterative training framework to achieve effective abnormal behavior detection in different surveillance videos. The design of the framework and the corresponding training process are described clearly. The evaluation is performed on estrus cow detection, and the performance (e.g., accuracy, mAP, and annotation effort) of the proposed approach is presented and analyzed. And the performance of the proposed approach looks promising.
Overall, the manuscript is well-organized and well-written. Readers of the journal can get valuable insights into the problem, the design of the solution, and the benefit of the solution.
Some minor suggestions are listed in the following for the authors to improve the manuscript.
1. In the beginning of the Introduction section, the detection of abnormal behaviors of humans is described first. However, the focus of the proposed approach is the detection of abnormal behaviors of animals. Based on current content in the first paragraph, readers might expect to see the detection of fighting, sneaking or so on of humans.
2. After line 73, the term "we" is highly used in the manuscript. It is suggested that the writing style can be checked again.
3. It is suggested that screenshots of training video, Stream 1 and Stream 2 used in the experiment can be presented. Thus, readers can get insights into the impact of background differences among videos, and the contribution brought by the proposed approach.
4. If more than one abnormal behavior should be detected and identified in the same time, can the proposed approach help? Any necessary modification?
1. After line 73, the term "we" is highly used in the manuscript. It is suggested that the writing style can be checked again.
2. Please proofread the manuscript if any modification is made after the revision.
Author Response
Comment 1: In the beginning of the Introduction section, the detection of abnormal behaviors of humans is described first. However, the focus of the proposed approach is the detection of abnormal behaviors of animals. Based on current content in the first paragraph, readers might expect to see the detection of fighting, sneaking or so on of humans.
Response: We reorganized the Introduction section to address detecting abnormal animal behaviors. In line 27, we added the focus of our proposed approach for the detection of abnormal behaviors of animals: “Abnormal behavior detection can be useful in detecting unusual behaviors of animals, such as the estrus habits of cows or cribbing of horses. In livestock farming, cattle estrus detection is the most crucial factor in monitoring cattle health and breeding management”. We also removed the confusing paragraphs that refer to the detection of fighting, sneaking of humans.
Comment 2: After line 73, the term "we" is highly used in the manuscript. It is suggested that the writing style can be checked again.
Response: We rephrased some sentences to make the manuscript is more readable to the readers.
Comment 3: It is suggested that screenshots of training video, Stream 1 and Stream 2 used in the experiment can be presented. Thus, readers can get insights into the impact of background differences among videos, and the contribution brought by the proposed approach.
Response: We have included visual examples in Figure 4, as suggested.
Comment 4: If more than one abnormal behavior should be detected and identified in the same time, can the proposed approach help? Any necessary modification?
Response: The base detection model inherits pretrained weights from the COCO Dataset and is then fine-tuned to adapt to different classes in a new environment. In this paper, the detection model is fine-tuned on estrus and non-estrus cow images. To further expand the detection model’s capabilities and enable it to identify more than one abnormal behavior, such as a “sitting” cow, we need to collect training images for this new class. These images should be included in the iterative training process, and the entire process, as described in the manuscript, should be repeated to achieve the desired accuracy. To handle the annotation process for multiple classes, we replace the binary classifier with a multiclass classifier in the iterative training process.
Comment 5: After line 73, the term "we" is highly used in the manuscript. It is suggested that the writing style can be checked again.
Response: We changed the writing style to make the revised manuscript more readable, as suggested.
Comment 6: Please proofread the manuscript if any modification is made after the revision.
Response: We proofread the revised manuscript, as suggested.
Round 2
Reviewer 4 Report
I would like you perform any statistic test (not only the values of acuracy and recall etc) to say that out proposal is statistically better than another
Author Response
Comment: I would like you perform any statistic test (not only the values of acuracy and recall etc) to say that out proposal is statistically better than another
Response: We conducted a paired t-test and added a paragraph in the revised manuscript, as suggested.